Journal of Data-centric Machine Learning Research (2024)        Submitted 10/24; Revised 03/24; Published 05/24

# Building Better Datasets: Seven Recommendations for Responsible Design from Dataset Creators

**Will Orr**  *University of Southern California, Annenberg School of Communication*
*Microsoft Research, New York*

**Kate Crawford**  *University of Southern California, Annenberg School of Communication*
*Microsoft Research, New York*

**Reviewed on OpenReview:** *https://openreview.net/forum?id=6bd8BrRKTW*

**Editor:** Yang Liu

## Abstract

The increasing demand for high-quality datasets in machine learning has raised concerns about the ethical and responsible creation of these datasets. Dataset creators play a crucial role in developing responsible practices, yet their perspectives and expertise have not yet been highlighted in the current literature. In this paper, we bridge this gap by presenting insights from a qualitative study that included interviewing 18 leading dataset creators about the current state of the field. We shed light on the challenges and considerations faced by dataset creators, and our findings underscore the potential for deeper collaboration, knowledge sharing, and collective development. Through a close analysis of their perspectives, we share seven central recommendations for improving responsible dataset creation, including issues such as data quality, documentation, privacy and consent, and how to mitigate potential harms from unintended use cases. By fostering critical reflection and sharing the experiences of dataset creators, we aim to promote responsible dataset creation practices and develop a nuanced understanding of this crucial but often undervalued aspect of machine learning research.

**Keywords:**  Responsible dataset creation, Ethical considerations, Data quality, Data documentation, Privacy, Consent, Machine learning, Interviews

## 1 Introduction

The acceleration of the development of machine learning systems has prompted a surge in the demand for high-quality datasets for training and evaluation. As demand for these datasets amplifies, the critical question is how to responsibly create and curate these collections. Building responsible machine learning systems requires making sure that datasets are produced in accordance with ethical considerations and responsible practices, as the use of unsuitable and inadequate datasets risks perpetuating societal inequalities and causing harm (Crawford, 2021; Keyes and Austin, 2022; Harvey, 2022). Therefore, intentionality and care in dataset creation are necessary (Famularo et al., 2021). Despite its importance, dataset work often receives less priority and funding than model design and algorithmic performance (Sambasivan et al., 2021). Dataset creators have been learning the lessons of this work, often in isolation, and their perspectives and experiences are rarely surfaced.

To address this gap, our research addresses the essential labor of creating datasets by presenting insights drawn from in-depth qualitative interviews with 18 leading dataset creators. In particular, we highlight actionable suggestions provided by practitioners for improving the responsible creation of datasets going forward. By giving prominence to their experiences, perspectives, and expertise, we aim to illuminate the current state of the field and draw attention to the challenges and considerations that dataset creators encounter in their work. The collective wisdom and experiences of these creators can beneficially direct the development of more responsible dataset production practices in the future.

This paper begins with an analysis of the existing literature on the production of datasets and reveals a lack of engagement with the voices of dataset creators themselves. Subsequently, we outline our methodology, comprising open-ended qualitative interviews with dataset creators from a wide range of research domains and organizational contexts. Despite their institutional differences, all participants expressed encountering common challenges of dataset creation such as striving for data quality, unexpected use cases of their dataset, and contending with issues of consent and privacy. Our findings highlight the need to overcome the current fragmentation and atomization within the dataset community, highlighting the potential for collaboration, knowledge sharing, and the cultivation of cross-sector expertise.

The central finding of this paper is that dataset creators share closely related concerns and suggestions; we share seven key recommendations for responsible dataset creation from dataset creators. These recommendations emerged in interviews as practical advice, rules of thumb, and lessons learned from failures. The recommendations include methods for attaining data quality, ensuring diversity within the dataset, learning from mistakes, and clearly communicating datasets' limitations and intended use cases. Processes for assessing the ethical concerns of datasets — including privacy, copyright, and consent — are also important open challenges. Sharing these experiences and expertise is essential to advance discussions on responsible dataset creation and foster a nuanced understanding of this crucial but often undervalued aspect of machine learning research.

## 2 Relevant Literature: The social construction of datasets

The creation of datasets is a crucial aspect of machine learning research, as they provide the foundation for both training and evaluating models. However, no dataset is a neutral, complete, or apolitical representation (Crawford and Paglen, 2019; Miceli et al., 2022). Practitioners must make a series of decisions throughout the dataset creation process (see Appendix 1) from data acquisition, labeling, cleaning, validation, and preparation of datasets (Roh et al., 2021; Polyzotis et al., 2018). And each of these practices is entangled with human conceptualization, judgment, and values (Gray and Suri, 2019). Decisions made during dataset creation involve inherently subjective determinations, assumptions, and contingencies which can have significant ramifications for the resulting dataset's biases and potential downstream harms (Jaton, 2021; Suresh and Guttag, 2021; Kang, 2023). However, once datasets are made available to the public, the contexts of their creation are lost, and these perspectives are frequently forgotten or ignored (Denton et al., 2020).

Frameworks for dataset development (e.g., Gebru et al., 2021; Luccioni et al., 2022; Famularo et al., 2021) are crucial for creating responsible machine learning systems. However, they do not provide insight into *how* datasets are made, nor the challenges faced by creators. Interventions such as datasheets (Gebru et al., 2021) data statements (Bender and Friedman, 2018), and nutrition labels (Holland et al., 2018; Chmielinski et al., 2022) contributes to safeguarding datasets by explicitly highlighting the practices of creation, intended uses and limitations. However, the modularity of dataset development (Polyzotis et al., 2018) can also hinder the thorough documentation of datasets, as no one individual can attest to each aspect of its creation (Widder and Nafus, 2023). A current limitation is that few research papers document how they constructed a dataset (Geiger et al., 2020).

The limitations of current practices for understanding dataset creation practices point to the need to facilitate the circulation of tacit knowledge of dataset creators. Despite recent calls to "bring the people back in" (Denton et al., 2020) to studies of datasets, the examination of datasets has yet to meaningfully include the voices of dataset creators and center their expertise on how responsible dataset creation can be improved. Understanding the insights, challenges, and expertise of dataset creators is necessary to comprehensively address the ethical implications of machine learning.

# 3 Methods

To conduct this study, we recruited "dataset creators" via email and conducted interviews between July and September 2022. We contacted 47 dataset creators with 18 respondents agreeing to participate (response rate of 38 percent). We use the term "dataset creator" to refer to individuals who possess substantial firsthand experience in creating datasets (Orr and Crawford, 2023). These participants were actively engaged in the development of their respective datasets, assuming various roles, both in junior and senior capacities. Many of these datasets were collaboratively produced, with participants contributing to specific aspects or overseeing the entire creation process. While we acknowledge the existence of other essential roles in dataset construction, such as data labelers (e.g., Gray and Suri, 2019), our primary focus in this context is on those responsible for compiling and circulating datasets as finalized products. Most of our participants were involved in multiple dataset projects, and their experiences from previous endeavors often influenced their responses.

Participants were identified by assessing the most cited contemporary datasets, and the dataset used to train and evaluate large proprietary models, such as DALL-E 2 and GPT-3. Some participants were contacted through personal connections or due to public interest in responsible dataset practices. Despite the prominence of some of these datasets, creators worked across a range of institutional and financial conditions, with many creators having to negotiate limited computational and financial resources (Orr and Crawford, 2023). Our 18 participants were located across the USA, UK, Europe, and Australia. The majority of participants (12) were employed by universities while creating their datasets; some worked for private corporations (2) and non-profit organizations (4). Semi-structured interviews traced the creation and circulation of datasets through their origins, usages, maintenance, and obsolescence. Participants were invited to reflect on what had worked well in the design of their dataset, challenges they encountered, practices they would change in hindsight, and general suggestions of best practices for future dataset creators (See Appendix 2 for relevant interview questions).

The datasets in question were also diverse, including natural language processing benchmarks, emotion detection, action recognition, personal recommendation, and large scraped multimedia corpora (see Appendix 3). Specifically, the datasets interrogated were: SQuAD 2.0 (Rajpurkar et al., 2018), GLUE (Wang et al., 2018), SuperGLUE (Wang et al., 2020), Words in Context (WiC) (Pilehvar and Camacho-Collados, 2019), IEMOCAP (Busso et al., 2008), YFCC100M (Thomee et al., 2016), Common Crawl (Common Crawl, 2023), C4 (Raffel et al., 2020), LAION 400M (Schuhmann et al., 2021) and 5B (Schuhmann et al., 2022), Amazon Reviews (McAuley et al., 2015), MovieLens (Harper and Konstan, 2015), WorldStrat (Cornebise et al., 2022), TweetEval (Barbieri et al., 2020), WinoGrad (Levesque et al., 2012), WinoGrande (Sakaguchi et al., 2021), UCF101 (Soomro et al., 2012), Taskonomy (Zamir et al., 2018) and IKEA Assembly (Ben-Shabat et al., 2023). The diversity of our sample was a strength of our methodology. We did not limit our sample to specific task communities, dataset collection methods, or training or benchmark datasets. Instead, we consider dataset creation as a holistic practice and present recommendations that may be relevant to all dataset creators, regardless of domain.

The interviews typically lasted around one hour, with durations ranging from 40 minutes to 1.5 hours. Participants did not receive compensation for their time. To ensure appropriate recognition and protect the anonymity of participants, dataset creators were given the choice of being identified in research outputs by their name, the datasets they created, or remaining anonymous. This research received approval from our organization's institutional review board. Interviews were transcribed and thematically coded iteratively, allowing new themes and recommendations to emerge as the interviews progressed.

## 4 Dataset creation as a fragmented field

Dataset creation is a fragmented domain characterized by a wide array of contributors. While some creators specialize in dataset development, others contribute to dataset creation incidentally. For McAuley dataset development is not "*a field as such. it's not like a full practice. It's just something people do*" [Interview, Amazon Reviews]. Some participants did not identify as a "*dataset person*" [Interview, C4] but rather perceived their work as menial, monotonous, and perfunctory, yet a necessary first step in the advancement of a new model for machine learning research. Moreover, those creators that do identify with dataset creation echoed Sambasivan et al. (2021) stating that "*data work is broadly undervalued in NLP and machine learning*" [Interview with Wang, GLUE; SuperGLUE]. This acknowledgment underscores a paradox whereby, despite the significance of dataset creation, it is overlooked and undervalued within ML research.

Despite differences in research domains, participants all faced common challenges including obtaining high-quality data, ensuring diversity and representation, and assessing if their datasets are discriminatory, or infringing privacy, copyright, the right to publicity, or other potential legal issues (Orr and Crawford, 2023). These challenges are often exacerbated by limited resources available, and the large scale of some datasets that hinder comprehensive auditing (Orr and Crawford, 2023). Participants highlighted that "*there's no real guidelines*" on how to address these shared and common challenges, having to instead implement their own ad hoc solutions [Interview with Oršolić, WorldStrat].

Amidst this fragmentation, we propose transforming dataset creation from "*just something that people do*" [Interview with McAuley, Amazon Reviews] to an established domain, with shared resources, tricks, and techniques to overcome collective problems. We support calls for the "professionalization of data work" (Famularo et al., 2021) with its own norms, tools, and habits. Studies with groups of experts across fields have demonstrated how exchanging experiences and stories, and leveraging collective expertise can foster collaboration, stimulate innovation, and provide social support for community members (Wenger, 2000; Wenger-Trayner and Wenger-Trayner, 2015). Participants echoed this sentiment and encouraged the community to "*have a broader look at how these things are done and get some sort of best practices in the field*" [Interview with Gould, IKEA]. Participants commended the development of the NeurIPS datasets and benchmark track for foregrounding dataset creation labor, and fostering collaboration and shared insights. In the next section, we will provide seven recommendations for responsible dataset creation drawn from interviews with dataset creators.

## 5 Practices for responsible dataset creation

The dataset creators we interviewed all provided valuable recommendations on how to develop datasets more responsibly. Practitioners emphasized the importance of striving for high-quality datasets; including diverse data collection, internal audits, data validation, curation, and iterative practices. Practitioners also noted the importance of considering the social, ethical, and epistemological implications of their creations. Each of these best practices was learned through their craft and lived experiences. Some of these recommendations had been learned through failure, with creators wishing to save others from similar experiences. Here we provide an overview of the seven most common recommendations.

### 5.1 Recommendation 1: Diversify your dataset and audit thoroughly

**Diversify your dataset**   Ensuring diversity and representation of various cohorts and conditions is a crucial aspect of responsible dataset creation. Dataset creators operationalize "diversity" in multiple ways, both regarding the demographics of participants and data subjects within the datasets and in terms of attributes within the data. Gould explained that "*Machine learning algorithms are lazy. They pick up on the simplest signals they can*" [Interview, IKEA]. Biases can infiltrate AI

systems during data collection and preparation (Suresh and Guttag, 2021), and when marginalized cohorts are underrepresented in training datasets, it can result in disparate impacts and societal harms (Buolamwini and Gebru, 2018). Additionally, confounding relationships in the data can lead to overfitting on specific test cases without addressing the underlying problem, thereby reducing the overall efficacy of the model (Srinivasan and Chander, 2021). Similarly, research highlights how diversity along all axes within a dataset can produce superior learning outcomes for models (Gong et al., 2019). Participants thus emphasize the significance of collecting diverse data across all aspects to mitigate dataset biases, overfitting, and "*spurious correlated signals*" [Interview with Gould, IKEA].

While some forms of diversity may be captured naturally by the data collection pipeline, others require intention and consideration from creators. The IKEA dataset, for example, was constructed by recording participants' movements while assembling furniture to capture natural variations in body postures. However, to create "a more holistic understanding" of pose detection, Gould [Interview, IKEA] explained they observed actors in five different environments. He continues, "*we wanted to have different environments. But we also wanted to make sure that the environment doesn't give away the action or activity being performed. And so that's why we have different furniture being assembled within the same environment. And different actors within the same environment.*" Intentionally collecting data across a range of factors, and their interrelations can mitigate "*spurious signals*" that may produce dataset biases.

Collecting data with sufficient diversity can require additional consideration, time, and resources. Despite their efforts, the IKEA team had difficulty recruiting women as actors. The actors were primarily students from their computer science department, and data collection reflected the demographic dynamics of the department: most participants were in their early 20s and only one quarter were women. However, the team did not have sufficient resources to continue "diversity-informed data collection" (Stasaski et al., 2020). Rather, the team settled for a dataset that underrepresented women. The unbalanced nature of the dataset had material consequences: an off-the-shelf human pose estimation algorithm performed worse on female participants in their dataset than on males. This was a concern for the team given the application of this dataset beyond the realm of furniture assembly, such as in robotic assembly arms on factory floors. The team hypothesized that clothing variations across genders, including headscarves worn by Iranian participants, may have contributed to this disparity, and the lack of diverse clothing present in the datasets used to train these models. This case underscores the critical role of diversity in dataset creation, the challenges associated with achieving it, and the potential downstream consequences of inadequate representation. Creators must consider these potential consequences when making datasets.

**Audit your dataset**    Given the importance of diversity in dataset design, creators also highlighted the centrality of auditing datasets to understand their distribution. Gould recommended that dataset creators "*slice the data up in different ways and report and be honest about the distribution of the data*" [Interview with Gould, IKEA]. For Jitsev, this meant involving a dedicated team "*who can produce sober and detailed statistics about datasets' content*" to ensure that "*the dataset is not going into the weirdly strongly non-balanced direction*" [Interview, LAION 400M and 5B]. Prior to distribution, an internal audit of a dataset can be used to find patterns and gaps, quantify undesired material and errors, and examine biases and societal consequences (Birhane and Prabhu, 2021). Audits can motivate additional steps to address imbalances in the initial dataset, such as purposefully sampling.

Auditing datasets also prompts creators to "*think about biases or think about who we are marginalizing*" by releasing datasets [Interview with Zamir, UCF101; Taskonomy]. Earnestly reporting the distribution of datasets across a range of characteristics may prevent the use of these datasets in cases where disparate impacts may be particularly harmful. Practitioners must be mindful that gathering data on marginalized groups to create "fairer" systems does not mitigate potential negative effects brought on by these systems (Miceli et al., 2022). In use cases that perpetuate historical marginalization and over-scrutiny, diversity-informed collection may give way to predatory inclusion.

While an important factor in responsible dataset creation, diversity is not the only factor in creating datasets responsibly.

## 5.2 Recommendation 2: Strive for higher dataset quality

**Understand and validate your data**   Striving for data quality was a key recommendation made by several participants. Quality was seen to include aspects such as accurately reflecting the target populations, ensuring no missing, inconsistent, or duplicate data, removing blurry, nonsensical, and inappropriate content, and maintaining relevance to the project objectives. High-quality data was seen to safeguard AI systems from erroneous or inappropriate outputs that may produce social harms or negatively affect those subjected to algorithmic decisions (Birhane and Prabhu, 2021; Wang and Strong, 1996). This requires thoroughly understanding the data in order to ensure its quality and usefulness. However, understanding large datasets thoroughly can be challenging, especially for large datasets that cannot be manually inspected in their entirety (Paullada et al., 2021). Validation is an important step in the creation of high-quality datasets, to ensure that the data represent the intended information. This requires "*testing, testing, and a lot of testing and validation to see what you're doing is correct*" [Interview with Oršolić, WorldStrat], and often practitioners rely on their own tests and rules of thumb. Polyzotis et al. (2018) propose "sanity checks" to verify that the data adheres to the expected properties prior to use. These include verifying the expected values of continuous variables, the distribution and modal values of categorical variables, and ensuring sufficient coverage for features across the dataset. Where the expected observations of datasets are unclear, creators can visualize the distribution of the dataset, which can uncover surprising properties about the data that may point towards systemic issues in the dataset creation pipeline (Polyzotis et al., 2018).

Manually inspecting the data is important to identify issues that may affect data quality. As Wang [Interview, GLUE; SuperGLUE] stated, it's important to "*look at the data, see what types of stuff are being asked for, see what the examples look like.*" Manual scrutiny can uncover unexpected data quality problems such as mislabeling, harmful or inappropriate content or labels, missing categories, or deeper epistemological issues. These checks are crucial to ensure synergy between the conceptualization and operationalization of constructs within the dataset (Blodgett et al., 2021). For example, by manually inspecting their dataset, Oršolić [Interview, WorldStrat] identified black bars in images due to metadata discrepancies between coordinate systems. His colleague thus recommended to "*take the time to dive into your dataset manually and get a feel for it*," which had been a "*massive debugger several times*" [Interview with Cornebise, WorldStrat]. Similarly, Bhagavatula [Interview, WinoGrande] urges practitioners to "*not blindly create a dataset*," but rather to inspect the data and ensure it represents what was intended. Furthermore, the publication of a dataset does not guarantee its proper validation or imply that it cannot be further improved; data validation and quality checks are also crucial for practitioners working with existing datasets and resources.

**Clean and curate your dataset**   For some creators, curating and cleaning datasets was imperative for ensuring the quality of the dataset. These practices include removing duplicate and low-quality content from the dataset, examples that may be incorrect or confusing, and content that may be inappropriate or capable of producing harm. While data curation is crucial it can also be difficult to achieve, as a C4 creator explains, "*If you want to feel confident in the dataset, then you need to curate it. And dataset curation is incredibly difficult and nuanced work*" [Interview, C4]. Creators noted not properly curating their datasets due to the immense human labor required to do so well.

Moreover, creators also noted the associated challenges of curating a dataset. In the case of WinoGrande, due to the presence of lots of similar and easy examples within their dataset, the team developed an algorithmic filtering pipeline that removed common and overrepresented examples. However, Bhagavatula [Interview, WinoGrande] explained that creating a curated, more challenging dataset came at the cost of data quality as their filtering algorithm "*tries to find examples that are hard in some sense, but noisy examples tend to be hard, because they are noise*". As such, Jia [Interview, SQuAD 2.0] underscored the importance of discipline in curation, to say "*I'm gonna make*

*the dataset really clean, but also challenging.*" Data curation is an imperative step in striving for data quality that must be approached with care and considerations for its consequences.

### 5.3 Recommendation 3: Start early and iterate.

**Learn from your mistakes**   Dataset creation is a process of immense complexity. Creators recognized the inevitability of mistakes and unexpected complications in their work and underscored the need to learn from their mistakes and adapt. In the case of the Words in Context (WiC) dataset, Camacho-Collados explained that they thought the creation process would be easier due to reusing existing resources and datasets. However, this process brought with it its own challenges: "*it was not a super smooth ride... we thought at the beginning it was going to be easier. But then there were a lot of considerations that probably we should have taken from the beginning.*" [Interview with Camacho-Collados, WiC; TweetEval]. Camacho-Collados thus emphasized the inevitability of unforeseen challenges: "*It's almost impossible to know from the beginning everything that can be expected with the dataset construction. You will always encounter some choices. Sometimes difficult choices, sometimes not so difficult. And then you have to go back. And then you can consider new things*" [Interview with Camacho-Collados, WiC; TweetEval].

Given the inevitability of unforeseen challenges, participants underscored the importance of learning from your mistakes and adapting to new conditions and challenges. Camacho-Collados recommended "*getting your hands dirty from the beginning*" as it helps in discovering the challenges that will inevitably arise during dataset creation [Interview with Camacho-Collados, WiC; TweetEval]. Camacho-Collados acknowledged that "*the things that you were thinking about in the beginning, maybe they were not so important and then you will find things that are more important.*" Dataset creators should accept that "*your first version of the dataset has almost zero chance that it's going to be the final dataset,*" and rather strive for continual improvement through "*manual checking*" and "*tinkering*" to ensure the best possible released dataset [Interview with Camacho-Collados, WiC; TweetEval].

**Iterate**   For creators working with crowd workers to collect and validate their data, practices of "*rapid iteration*" were particularly important [Interview with Bhagavatula, WinoGrande]. Creators emphasized the importance of honing the categories within the datasets, the instructions for crowd workers, and the criteria for selecting workers. Gould [Interview, IKEA] explained that when creating instructions for crowd workers, "*You're not going to get it right in the first instance. So, you sort of want to iterate and visualize as you go, track as you go, and then improve the process as you go*" [Interview with Gould, IKEA]. These processes of identifying errors, learning from mistakes, and making iterative improvements to the dataset creation pipeline are crucial for creating high-quality datasets.

### 5.4 Recommendation 4: Document datasets openly and communicate limitations

**Communicate limitations**   Several creators underscored the importance of thoroughly documenting the processes of dataset development, and communicating the strengths and weaknesses of datasets for specific contexts and objectives. However, some participants felt that the academic peer review system discourages dataset creators from being reflexive and thoroughly presenting the limitations of their datasets. Zamir explained the perception that if creators detail the flaws of their datasets in their papers, "*You are going to get rejected more often because the reviewers would just look at limitations and copy-paste into a review*" [Interview, UCF101; Taskonomy]. This has created a dataset "documentation debt" in which prominent datasets are used and circulated without records of their contents or limitations (Bandy and Vincent, 2021).

Despite potential pushback, creators like Zamir emphasized the need to "*in an explicit way, discuss limitations*" of datasets "*even if that happens in retrospect*" [Interview, UCF101; Taskonomy]. Notably, the datasheets for the datasets framework (Gebru et al., 2021) had been adopted by several

of the creators we interviewed (e.g., Cornebise et al., 2022; Schuhmann et al., 2022), who commended its integration within NeurIPS. Creators recognized the potential of such frameworks for broader application and encouraged other conferences to take a similar approach.

**Document Continuously**   Documentation practices should not stop with the release of a dataset. Creators underscored the need to continue to document datasets as new limitations arise and as the dataset becomes obsolete. Over time, all datasets become considered obsolete by both the creators themselves and the broader field of machine learning. Newer, superior, or more challenging datasets may be released (e.g., Yang et al., 2020), and models may achieve near-perfect prediction scores, rendering the dataset "solved" (e.g., Baldominos et al., 2019). In some cases, dataset creators may also retract datasets due to the identification of harmful, privacy-violating, or inappropriate content (e.g., Torralba et al., 2008).

However, standardized practices do not yet exist for retiring datasets (Luccioni et al., 2022). Zamir observed that datasets are rarely retired voluntarily, "*they happened because they had to be taken down because of copyright issues or because of inappropriate content and so on* [Interview, UCF101; Taskonomy]" Establishing clearer standards for retiring datasets is essential to curtail the misappropriation and misuse of outdated, flawed, or inappropriate datasets (Conference on Neural Information Processing Systems, 2023). For Zamir, dataset creators could have labels or headers on dataset distribution websites to communicate new limitations as they are identified and suggest more suitable datasets as they are released [Interview, UCF101; Taskonomy]. This may foster a cultural shift against using outdated and flawed datasets. However, more work needs to be done to prevent their circulation on third-party platforms like Academic Torrents (Harvey, 2022). The inclusion of datasets on third-party distribution sites solidifies certain versions of the dataset and it becomes difficult for the original creators to exercise control over their work. This includes updating the dataset to reflect new findings or corrections, maintaining its quality and relevance over time, or even removing it if necessary, for instance, due to ethical concerns or data privacy issues (Luccioni et al., 2021).

**Ensure proper attribution**   Attribution was also an important consideration for dataset creators, particularly those using existing datasets and resources. Camacho-Collados explained that TweetEval, a meta-benchmark, was uploaded to Hugging Face, a dataset library, by a community member [Interview, WiC; TweetEval]. In the process, attribution to the original creators was removed, resulting in confusion and concern from the original creators. GLUE, another meta-benchmark, faced a similar challenge of ensuring the continued attribution to the original creators of the datasets. As GLUE creator Wang explained, "*What we ultimately settled on is to ask people who use the dataset to cite the original papers. And when we saw papers that didn't do that, to the best of our abilities would message authors and ask them to do that*" [Interview, GLUE, SuperGLUE]. Creators, therefore, highlight the need for clearly communicating licensing and terms of use restrictions for users of a dataset and ensuring that these are followed.

**Strive for Reproducibility**   Documenting datasets can also benefit creators. Unforeseen errors can hinder dataset creation when they are identified, requiring creators to start over, as Camacho-Collados [Interview, WiC; Taskonomy] reflects: "*we worked a lot and then suddenly had to start again and it was not so easy to recreate the process*". To combat this, the WorldStrat team decided to ensure that their dataset was entirely reproducible from the outset, by releasing both the dataset and the code to reproduce the dataset. This approach "*saved [their] butt a few times*" as it allowed the dataset to be easily salvaged and corrected as errors were identified [Interview with Oršolić, WorldStrat]. As Oršolić [Interview, WorldStrat] explains, "*it's so much easier to just rerun a notebook and it's going to generate everything for you and knowing that anybody else can do the same.*" Thorough documentation thus promotes the careful and responsible creation and modification of datasets.

**Encourage user engagement**   Open and thorough documentation also encourages users to use the dataset and modify it if errors are identified. Camacho-Collados [Interview, WiC; TweetEval]

explained that while he "*wanted to have everything perfect for releasing... it's never going to be the case.*" Flaws and errors, however minor, are almost always identified after the release of datasets. However, he explained that due to releasing WiC openly with sufficient documentation, other teams were able to adopt the project and extend it in various ways. Similarly, Cornebise [Interview, WorldStrat] highlighted this benefit of dataset documentation in promoting data reuse and improvement: "*I really want to see more people using the way we've built it and the thinking and improving on what we've done. That's why we're very careful to put a long datasheet for the dataset.*" Documenting datasets thus not only arms users with the necessary information to decide when and how to use a dataset, but it can also encourage the creation of higher-quality datasets in the future.

### 5.5 Recommendation 5: Create user-centric datasets and limit inappropriate applications

**Define intended use cases and user groups**   Clearly defining and communicating the intended use cases of datasets to potential users was of paramount importance to dataset creators. Participants cautioned against creating datasets for the sake of it. In their view, datasets should "*fill up a gap*" [Interview, Camacho-Collados, WiC; TweetEval] in a particular domain or task. Clearly articulating these use cases is crucial as they shape the dataset and its suitability for downstream tasks. Narayanan [Interview, IEMOCAP] explained that datasets are "*not designed for the world to use necessarily beyond what the researcher intended*". While he acknowledges that the reuse of datasets may inspire meaningful and responsible innovations, using datasets in unintended contexts can incur inherent costs and limitations. Indeed, datasets are frequently adopted by external communities for purposes beyond their original use cases (Koch et al., 2021). Clearly communicating these intended use cases may help prevent the unsuitable use of datasets.

Clear use cases also assist creators in tailoring their datasets to meet the needs of their intended user groups. As Nagel [Interview, Common Crawl] suggests, "*think about your user group and how you would design your dataset specifically for the user group*". For example, the WorldStrat team prioritized enabling NGOs with limited funding to use their dataset by collecting most of the data from freely available sources and ensuring compatibility with modest computer systems. Other creators aimed to create user-centric datasets by making them as "*simple as possible*" [Interview, Camacho-Collados, WiC; TweetEval] to ensure usability for a wide audience with varying technical skills. As Camacho-Collados [Interview, WiC; TweetEval] explains, "*not everybody has a computer science degree or they are not professional coders or work on NLP or Machine Learning. Other people working in other fields, they are very interested in these kinds of things as well.*" Tools such as browsers, visualizations, and websites can facilitate data auditing and understanding to ensure accessibility for all users. This drive for technological accessibility aligns with calls to "democratize" AI technology development and utilization (Seger et al., 2023). Some creators also suggested creating "*specialized subsets*" [Interview, YFCC] of datasets targeting specific use cases and communities, each with their own unique purposes, advantages, and challenges.

**Anticipate unintended use cases and harms**   Despite some creators having clear use cases, creators all expressed that their datasets had been taken up in ways that they had not expected. For example, some participants noted that their datasets intended to evaluate the capabilities of language models were used to evaluate language generation. Moreover, one dataset of product reviews intended to be used for recommendation systems had been taken up within sentiment analysis research. These cases illustrate the ways in which datasets may flow between task communities (Koch et al., 2021), which may require additional strategizing about potential impacts. Furthermore, some creators never intended to create datasets for AI applications. For Narayanan [Interview, IEMOCAP], the dataset "*was a byproduct of a specific research project or research experiment that has been since broadly used.*" Some creators thus felt powerless to control the downstream impacts and potential misuse of their dataset. To address these concerns, creators recommended carefully considering all potential

use cases and impacts during dataset development, including the possibility of undesirable or harmful uses.

While the peer review and the publication pipeline provide important checks and balances, creators can take preventative measures and address these limitations and harms. This includes identifying and removing potentially harmful content that could be misappropriated. Once potential harms are identified, creators must ask, "*How do you either prevent it if you can, or warn against it if you can't prevent it?*" [Interview with Cornebise, WorldStrat]. In cases where prevention is not feasible, creators may need to make the difficult decision of whether to release the dataset at all. For instance, in a previous project, Cornebise [Interview, WorldStrat] developed a dataset of satellite imagery of displaced populations, refugee camps, and destroyed villages in Darfur, Sudan. However, this dataset was never released due to the potential for misuse. He explained, "*we didn't want to risk painting a target on the villages because there's no map of Darfur. So putting them out there could potentially have a risk there. What if the tools that we develop get used by a military or a terrorist organization?*" By taking steps to mitigate potential negative consequences and limiting harmful uses where possible, creators may be able to prevent deleterious applications of their creations – but they also need the courage to stop a dataset release when necessary.

### 5.6 Recommendation 6: Contend with privacy and consent

**Consider privacy implications** Most creators encountered privacy and consent concerns during dataset construction. In some cases, these challenges arose after the release of the dataset, leading to criticism and requests for the removal of personal data. While creators were always sensitive to the legal constraints of their work, they recognized the inadequacy of current legal frameworks for addressing privacy and consent concerns (see Birhane et al., 2021). Thus dataset creators are encouraged to consider privacy beyond their legal obligations. In the case of YFCC, a large-scale image dataset scraped from Flickr, the violation of data subjects' privacy had flow-on effects: YFCC serves as the basis for numerous derivative datasets, including the controversial MegaFace facial recognition dataset used by companies, governments, and law enforcement to improve facial recognition and surveillance technologies (Harvey, 2022). Reflecting on these concerns, a YFCC creator highlighted the need to consider privacy as an end in itself rather than a legal constraint to be met: "*With the knowledge today, the privacy concerns, we probably should have addressed them. We only addressed them legally. But not on a reflective element... Because we didn't know how successful this dataset might be. This is staying there forever.*"

Some creators took measures to retain data subject autonomy in the creation of datasets. In the case of data collection through scraping existing web sources, participants underscored the importance of having "relatively polite settings when crawling" [Interview with Nagel, Common Crawl]. Crucially, this includes adhering to existing standards of consent such as following robots.txt rules to avoid the collection of data from sites that explicitly block crawlers. This also entails clearly identifying the crawler to provide transparency about its purpose and origin. Limiting the rate at which requests are made to a website is also an important step towards polite scraping to prevent overwhelming the site's server. Furthermore, avoiding crawling every page unnecessarily (if the objective can be achieved with a smaller subset of pages) can mitigate unnecessary burden on website administrators.

Moreover, some creators implemented data removal requests allowing for data subjects to request the deletion of their data from the dataset. Other creators only provided links to the data (such as image files), rather than the data itself as this allows subjects the ability to remove their content. However, both of these measures prioritize data subject autonomy only after their data has been collected and circulated within datasets. While tools such as *Have I Been Trained?* (Spawning) are useful for data subjects to assess whether they have been included within datasets and facilitate opting out of datasets, the burden of opting out still falls on the data subject, rather than a consent-forward model that enables them to opt into a dataset. Creators also pointed towards technical solutions to safeguard data subject privacy such as transformation techniques that create synthetic data

that removes identifiable information. However, these solutions are still in development. As Jitsev [Interview, LAION 400M; 5B] states, "*there is still a lot of research required on how to solve such problems.*"

**Improve consent measures**  Creators expressed the need to improve methods for obtaining consent within datasets. As yet, there are no clear ways to obtain consent for scraped data included in ML datasets. Nor are there established means to remove personal information from datasets that have already been downloaded, derivative datasets, or versions that have been distributed on third-party platforms. Currently, the ability for data to be accessed publicly is taken as consent for the data to be used in any way deemed suitable, including as training data within machine learning models (Birhane et al., 2021). Jitsev [Interview, LAION 400M; 5B) reiterated this sentiment: "*whatever you crawl publicly is publicly available. So all these things that are ending up in datasets are already in there. They just need a tool to be accessed.*" However, this assumption must be reevaluated. Other creators expressed the limits of this model of consent, as a C4 creator explains, "*the fact that someone types something into the computer and posts it publicly on the Internet doesn't necessarily mean that they intended for it to be used to train a language model, and possibly be memorized by a language model in perpetuity*" [Interview, C4]. Clearer licenses pertaining to machine learning systems are thus needed. Creators encouraged the field to reflect on the lack of meaningful consent when being included in datasets and the potential downstream implications.

### 5.7 Recommendation 7: Make the datasets you need

**Create fit-for-purpose datasets**  Dataset creators understand the importance of developing datasets that are fit for purpose in ML research. While creators acknowledge the constraints involved in dataset development such as time, money, and access to computational resources, they emphasize the benefits of creating datasets tailored to their specific needs. Relying on "found" data to solve computational problems is a common but limiting practice. As Wang notes, "*people largely just hope to find naturally occurring data that they can turn into a dataset that they like, rather than trying to create the data that they want*" [Interview, GLUE; SuperGLUE].

Participants encouraged people to create their own datasets that suit their specific research needs. For example, Harper [Interview, MovieLens] notes that "*too many researchers rely on the existence of online datasets for their research, and too few create their own datasets.*" Similarly, Wang [Interview, GLUE; SuperGLUE] believes researchers should be "*more willing to work with human annotators*" to create the specific datasets necessary for their project needs. Creators acknowledge the difficulty of dataset development work and the constraints that may discourage outsiders from creating their own datasets, such as human, computational, and financial resource requirements. These constraints were particularly felt by creators of image, video, and multimodal datasets. Nonetheless, they encourage researchers to develop datasets that suit their specific research needs, rather than relying on existing scraped data outside of its intended context, or ill-suited proxy datasets.

## 6 Conclusion

Dataset creators are central to the process of strengthening responsible machine learning practices, yet their perspectives have been underexplored in the existing literature. This paper has sought to bridge this divide by offering insights gleaned from interviews with 18 preeminent dataset creators so the field can learn from their challenges and deliberations. Particularly, we see opportunities for enhanced collaboration, knowledge exchange, and shared growth in the dataset community. In our analysis of the views of these creators, we share seven pivotal recommendations to bolster responsible dataset creation, including issues ranging from data quality and documentation to privacy, consent, and harm mitigation in unforeseen use scenarios. While each dataset will face its own ethical and epistemological challenges, these recommendations can serve as a starting point for dataset creators to

safeguard their datasets and limit the potential for serious harm. By encouraging critical introspection and disseminating the insights of practitioners, we argue for grounding responsible machine learning in the realities of dataset creation. Dataset creators have much to offer to improve the technical and ethical dimensions of this pivotal, yet often overlooked, domain of machine learning.

## 7 Limitations

A notable limitation of our research is that while our corpus of interviews represents a range of domains, it is by no means representative of the scope of dataset creation. Our sample is skewed towards highly cited datasets. Specific task communities or niche domains may not be captured by our sample. These Specific domains may encounter unique challenges that have not been captured by our recommendations. For instance, datasets intended for use in medical contexts may require additional considerations, such as heightened privacy risks. Domain-specific challenges and recommendations for dataset creation present excellent considerations for ongoing research. The creation and utilization of synthetic datasets is also an emerging area that is unexplored within the scope of this research. Given their nascent stage while the interviews were being conducted and the unique challenges that creators of synthetic datasets may face, they were not discussed with interview participants. Additional research is needed to examine the unique challenges, opportunities, and best practices associated with synthetic dataset creation.

Moreover, our research predominantly focuses on publicly available datasets created at academic institutions. The perspectives of creators of proprietary datasets in corporate contexts are largely absent from our research, and these creators may face unique challenges. This is the subject of our forthcoming work. Furthermore, while our research includes the perspectives of some creators of community-built datasets (e.g., LAION 4M and 5B), the unique challenges faced by these creators deserve more research in their own right. We also must note that all participants we interviewed were situated in Western countries. This is likely a consequence of our sampling process, which prioritized highly cited datasets and those that have been used in (English language) proprietary models. As Koch et al. (2021) note, these datasets tend to originate from a small number of elite institutions in Western countries. Further research is necessary to examine these issues in non-Western settings.

Finally we also acknowledge the importance of rigorous evaluation and adaptation of these guidelines to specific contexts and challenges. While the purpose of this paper is to present recommendations for dataset creation as they were communicated by participants, how these recommendations may be evaluated is beyond the scope of this paper, and is the subject of forthcoming work. We encourage users to examine the documentation of datasets and use these recommendations to think critically about the steps taken to produce and circulate datasets (see also Appendix 1). Further, we encourage ongoing dialogue and collaboration among dataset creators, users, and researchers to collectively advance our understanding and implementation of responsible dataset creation practices.

# 8 Broader Impact Statement

The responsible creation and curation of datasets is of paramount importance for machine learning research. Our work is driven by a deep commitment to illuminating the critical, yet often overlooked, aspects of dataset creation using qualitative approaches. We believe that by sharing the insights and experiences of dataset creators, our research can have a significant impact on the field, steering it towards a more ethical, responsible, and collectively informed future.

Our research aims to bring together the dispersed voices and experiences of dataset creators, a community that often operates in isolation. We anticipate that one of the key impacts of our work will be fostering collaboration and knowledge sharing among dataset creators. By highlighting the shared challenges, practices, and recommendations from our interviewees, we hope to contribute to transforming dataset creation from a fragmented, solitary endeavor into a recognized domain with its own norms, tools, and best practices. Dataset creators have learned valuable lessons through their experiences, successes, and failures. Our work amplifies the importance of these lessons and insights to ensure that they are not lost or ignored. We hope that dataset creators across domains will find common ground and that our findings will serve as a catalyst for a more cohesive and supportive community.

One of the central outcomes of our work is the seven practical recommendations for responsible dataset creation drawn from the insights of dataset creators. These recommendations encompass a wide array of topics, from striving for data quality and diversity to considering ethical and legal implications. While we acknowledge that no set of recommendations can ever be complete, and nor can it guarantee that a dataset is universally "responsible" due to varying domain-specific and data collection method requirements, we believe that following these best practices is a significant step towards more responsible dataset creation.

It is crucial to note that the responsibility of a dataset does not automatically imply that the use cases to which it is applied are inherently responsible. Users of datasets must reflect upon the context and purpose of their use case and the uses intended by the creators of the dataset to determine whether their specific use aligns with responsible practices. Our work encourages dataset creators and users alike to be rigorous in their assessment of use cases, ensuring they uphold ethical and responsible principles throughout the lifecycle of a dataset.

Our work underscores the importance of not only adopting best practices but also continuing to collaborate and adapt to the evolving landscape of dataset creation. We recognize that responsible dataset creation is an evolving field. Therefore, we emphasize the need to develop best practices that are continually refined through shared insights.

# 9 Acknowledgements and Disclosure of Funding

This work began as an internship project at the Fairness, Accountability, Transparency and Ethics group at Microsoft Research, New York. We would like to express our sincere gratitude to our colleagues and fellow interns at Microsoft Research, and all the members of the Knowing Machines research project for their generous feedback and support throughout the development of this work. We would also like to thank Arjun Subramonian for their generative feedback on drafts of this paper.

## Appendix 1: Dataset creation process

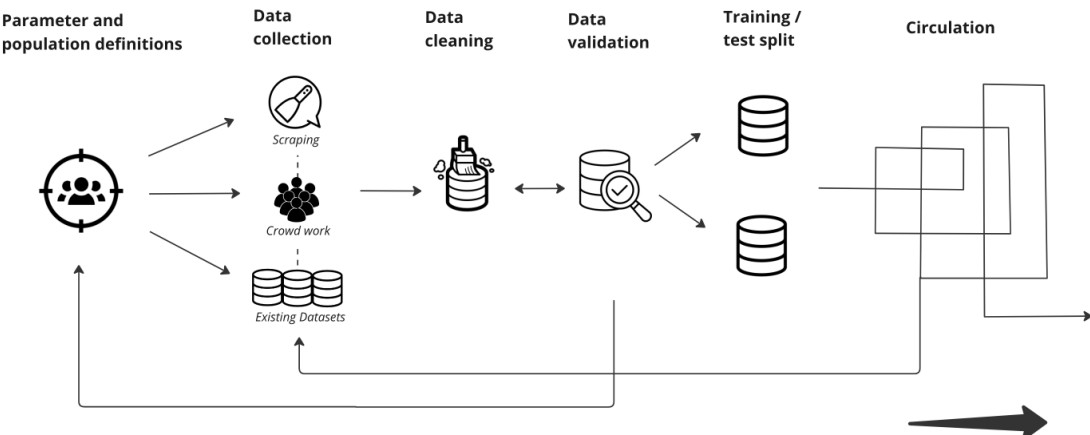

Figure 1: Flow chart of dataset creation from conception to circulation

**Parameter and population definitions**   This phase involves identifying the specific variables of interest within the broader group from which data will be collected. This phase is crucial for defining the phenomena the dataset intends to capture, what is within scope, and what might be considered inappropriate, invalid, or irrelevant data.

**Data collection**   Data is gathered from various sources according to the defined parameters and population. This can involve different methodologies such as scraping content from the web, employing crowd workers, or utilizing and repurposing existing datasets. Methods used are often shaped by the intended objectives of the dataset, as well as operational constraints, such as institutional requirements or resource limitations. Creators may utilize multiple data collection methods (such as employing crowd workers to label scraped content).

**Data cleaning**   Once data is collected, it often contains errors, duplicates, or missing values that need to be addressed. Data cleaning involves processing the data to correct inaccuracies, remove irrelevant information, and handle missing data.

**Data validation**   This phase checks the dataset for accuracy and consistency with real-world phenomena. Data validation ensures that the dataset meets the necessary standards and assumptions for its intended use. If these processes are deemed insufficient, creators may return to the data cleaning to improve data quality, or rethink how parameters are defined, to ensure the data captures what is intended.

**Training and test split**   In some cases, the dataset is divided into training and test sets. The training set is used to build and train the model, while the test set is used to evaluate its performance and generalizability to unseen data.

**Circulation**   After the dataset has been created, cleaned, validated, and split, it is made available for use by others. Circulation can involve publishing the dataset in public repositories, within an organization, or as part of a research paper, allowing other researchers or practitioners to use the data for their own analysis or model training. Datasets may find their way onto third-party distribution sites outside creators' control. They may also be taken up by researchers to create derivative datasets.

## Appendix 2: Relevant Interview Questions

The following interview questions were used to guide data collection. They have been categorized below to identify the broader purpose of the question. While each of these broader topics was examined in each interview, these questions were used as an entry point to understand the participants' perspectives, which were then developed through follow-up questions. Interviews were semi-structured in nature allowing the interviewer to remain open and flexible and hone in on responses as they related to the research (Warren, 2001).

**Motivations and objectives**

- What were you hoping to achieve with this dataset?

**Dataset creation processes**

- What was your role in the design of this dataset?

- Can you talk me through how you collected the data? How did you decide on the particular data sources?

- Can you talk me through how this data was labeled? (if applicable)

- How was the quality of the data evaluated? What were the processes of cleaning and validating the dataset?

**Challenges of dataset creation**

- Thinking back to some of the team's discussions, were there any moments that stand out to you as needing extra discussion or strategizing?

- Any additional challenges in creating the dataset?

**Reflections, improvements and recommendations**

- Are there ways the dataset has been used that have surprised you?

- Have you received any requests for changes or criticisms of the dataset?

- Reflecting on your dataset as a whole, did you achieve what you were hoping to?

- Are there any criticisms you would make of your own process? Looking back, what would you have done differently?

- What suggestions would you make to designers of datasets in the future?

# Appendix 3: The datasets

| Dataset | Domain | Data collection | Scale |
|---|---|---|---|
| TweetEval (Barbieri et al., 2020) | NLP meta-benchmark | Existing datasets | 200k (Across 7 datasets) |
| IKEA Assembly (Ben-Shabat et al., 2023) | Action recognition | Actors and crowd work | 16k |
| IEMOCAP (Busso et al., 2008) | Emotion detection | Actors | 10k |
| Common Crawl (Common Crawl, 2023) | Text corpus | Scraped | 1.4t tokens monthly (est. from April 2019 crawl) |
| WorldStrat (Cornebise et al., 2022) | Image detection | Existing datasets | 3.4k |
| MovieLens (Harper and Konstan, 2015) | Personal recommendation | User-generated | 100k, 1m, 20m and 25m |
| WinoGrad (Levesque et al., 2012) | NLP benchmark | Expert-generated | 285 |
| Amazon Reviews (McAuley et al., 2015) | Personal recommendation | Scraped | 180m |
| Words in Context (Pilehvar et al., 2019) | NLP benchmark | Existing datasets | 7.5k |
| C4 (Raffel et al., 2020) | Text corpus | Existing datasets | 156b tokens |
| SQuAD 2.0 (Rajpurkar et al., 2018) | NLP benchmark | Scraped and crowd work | 44k |
| WinoGrande (Sakaguchi et al., 2021) | NLP benchmark | Crowdwork | 108k |
| LAION (Schuhmann et al., 2022) | Image-text corpus | Scraped | 5.85b |
| UCF101 (Soomro et al., 2012) | Action recognition | Scraped | 13k |
| YFCC100M (Thomee et al., 2016) | Image detection | Scraped | 100m |
| GLUE (Wang et al., 2018) | NLP meta-benchmark | Existing datasets | 196k (Across 11 datasets) |
| SuperGLUE (Wang et al., 2020) | NLP meta-benchmark | Existing datasets | 1.485m (Across 12 datasets) |
| Taskonomy (Zamir et al., 2018) | Depth and surface estimate | Existing datasets | 4.5m |

Table 1: Datasets covered in interviews

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
