# OpenReview forum: "Building Better Datasets: Seven Recommendations for Responsible Design from Dataset Creators"
_DMLR — Accepted by DMLR_

### Review · Reviewer_4fB1 · 2024-01-16

**Recommendation:** 3
**Confidence:** 2

**Summary Of Contributions:**

The paper interviews 18 dataset creators to study the challenges and best practices of creating datasets responsibly. The recommendations cover the areas of data quality, documentation, privacy, harm mitigation, etc.

**Strengths:**

1. Relevant topic to the audience
2. The datasets whose creators are interviewed are mostly well-known, so the content is worth knowing

**Broader Impact Concerns:**

See Limitations.

**Claims And Evidence:**

The evidence is strong. Claims on the "responsible" part is little weak, see Limitation 2.

**Datasets And Benchmarks:**

No obvious problems for a survey-based paper.

**Extended Submissions:**

N/A.

**Limitations:**

1. The study seems to diverge from conventional survey-based HCI studies. It does not have research questions (RQs) laid out in the beginning, and how those RQs impact the design of the survey question. In fact, there isn't any explanation of the rationale for how the questions were designed. I suggest authors to discuss it before getting into the recommendations.
2. I find the *responsible* design part of dataset creation is less relevant than the paper claims in the title and introduction. Many of the recommendations are about data quality, e.g. recommendations 2, 3, 7. Normally, I would expect they are mostly about conventional responsible AI principles, e.g. fairness, privacy, safety, transparency, etc. I find the theme of the paper is not as well-supported as claimed.
3. It would be beneficial for readers to add limitations and discussions on the impact of different types of datasets. For example, the responsible design of creating NLP data could differ from recommendation data in many ways. The paper's recommendations are quite general. This is not asking authors to rewrite heavily, but adding a discussion/limitation to avoid giving readers a false sense of generalizability of recommended principles.
4. Although admitted in the limitation, the interviewers are biased toward academia (12 out of 18), and some recommendations seem to be unlikely to generalize to industry, e.g. "The actors were primarily students from their computer science department, and data collection reflected the demographic dynamics of the department".
5. Why is Section 4 (Dataset creation as a fragmented field) an independent section? I find the structure a little strange. If this point is very important to be a standalone section, or necessarily be mentioned first, I think it should be motivated accordingly.

**Requested Changes:**

See Limitations.


Overall, I find the paper can attract the potential audience because the topic is relevant, and there are no major flaws in the study as far as I can see. However, I personally find the takeaways are already known in the field and I do not find anything substantially new. But this is just my personal view. If the editor thinks this is a good summarization and study of useful principles, then I would recommend to accept with revisions.

---

### Review · Reviewer_zrEc · 2024-02-26

**Recommendation:** 4
**Confidence:** 3

**Summary Of Contributions:**

This paper presents a study of dataset development from the perspective of the dataset creators. From a series of interviews with a sample of practitioners who have created widely-adopted datasets, the authors pull out 7 themes of recommendations that are aimed at providing a roadmap towards responsible dataset design. This roadmap is especially important given the often ad hoc nature of dataset development across the field.

**Strengths:**

(a) Datasets are a central driver of progress in ML, but there is little work describing what makes a dataset “good” or even appropriate for a given use. This paper fills an important gap in outlining, from the dataset creators’ perspective, how they view their role in this process.

(b) The recommendations are clearly called out, and this paper can be read as a guide for ensuring standards of dataset creation. The authors are careful to note where recommendations will be more or less applicable to a given dataset’s needs, and they provide non-judgmental concrete examples of their recommendations throughout.

(c) The set of recommendations is thorough, and touches on each step in the dataset creation process (and after the dataset has been released).

**Claims And Evidence:**

All recommendations are supported with quotes and examples from the interview process.

**Datasets And Benchmarks:**

None

**Extended Submissions:**

Not applicable

**Limitations:**

(a) The datasets covered, though diverse, seem to be missing two broad categories: (i) datasets for specific science applications (e.g., agriculture), and (ii) synthetic datasets. Were these out of scope? Or were authors of these kinds of datasets contacted and they didn’t end up contributing? I’m curious if there would be additional recommendations that would need to be taken into consideration for these types of datasets. For example, as models trained on medical imaging datasets are actually used in diagnosing people, do the underlying datasets require additional work that, say, an NLI benchmark doesn’t? For synthetic datasets, how are the expectations around reproducibility different from other kinds of datasets?

(b) The dataset creators interviewed are located in Western countries. The authors should call this out as a limitation and address why this is the case.

(c) Despite the fact that many of the participants did not view themselves as being a “dataset person,” (section 4) there *are* many people in the field who define themselves this way, so I’m wondering (i) how the findings might be different if more people from this population were included, and (ii) whether the datasets created by people who do think of themselves this way end up being less impactful, which would have led to them being overlooked for this study.

(d) The authors do not take a strong stand in how dataset reporting should happen. They mention several existing efforts around dataset documentation (e.g., datasheets), but it’s not clear how all of the recommendations map onto this. If issues around the diversity of a dataset are addressed in idiosyncratic ways by each developer, that’s still better than not auditing, but is less useful than a set of standards around this. Based on the interviews with dataset creators, I’m wondering if they had a stand on what reporting best fit their needs as dataset developers, and then if that reporting fits the needs of people who will use the dataset.

(e) Some of these recommendations like ‘understand and validate your data’ require steps that cannot be verified by another individual. As the broader impacts section indicates that this study is intended not only for dataset creators, but also users of datasets who don’t know what to look for in identifying a good dataset, how should those individuals use these recommendations when they’re assessing dataset quality?

**Requested Changes:**

**Major:**
- (i) Address limitations (a) and (b) in the limitations section
- (ii) Limitation (d) may require an additional section to fully address. But it also seems that the authors may have been intentional in *not* calling out exactly how these recommendations should be ensured. If this is the case, it would be helpful for them to indicate why that is.
- (iii) Limitation (e) can be addressed in a few places where there may be a recommendation that isn’t possible to audit, but should be audited. I believe that the following are such recommendations: “understand and validate your data”, “clean and curate your dataset”, “iterate”.

**Minor:**
- (i) How many of the participants chose to remain anonymous? (bottom of page 3)
- (ii) Figure 1 caption is missing
- (iii) Recommendation to add a column to Table 1 noting if the dataset is for training, finetuning, or evaluation.
- (iv) Some discussion of limitation (c) would be useful, but I suspect that I’m just asking the authors to speculate about something they may not have insights on from this study, so I’m including it as a minor point.
- (v) I found the point on page 8 about Academic Torrents to come out of the blue. Could the authors clarify what the risks are here? I didn’t understand this point.

---

### Review · Reviewer_H5e4 · 2024-03-03

**Recommendation:** 4
**Confidence:** 3

**Summary Of Contributions:**

The paper dives deeply into the challenges of creating responsible and ethical datasets grounded in the experiences of dataset creators. It critically assesses the current landscape of dataset development, emphasizing the need for a paradigm shift towards more ethical practices. The authors meticulously dissect the dataset creation process, proposing seven actionable recommendations to foster ethical considerations, inclusivity, and diversity in dataset creation. These recommendations serve not only as a guideline for current creators but also as a foundational framework for future research and development in data-centric machine learning research.

**Strengths:**

The main strengths can be summarized as follows:

* **Significance and novelty:** The study bridges a gap in the literature by focusing on dataset creators' perspectives, contributing significantly to the discourse on ethical dataset creation.

* **New data collection insights:** The recommendations summarized via interviewing 18 leading dataset creators are highly relevant to the broader research community, emphasizing the importance of diverse, quality data and ethical considerations in dataset creation.

* **Research quality:** The qualitative analysis, involving in-depth interviews with experienced dataset creators, ensures high research quality and credibility of findings.

* **Clarity:** The paper is well-organized, presenting complex issues in an accessible manner.

**Broader Impact Concerns:**

The broader impact is well discussed in the paper. While the paper discusses ethical considerations in dataset creation, it would be better if authors could further explore the broader societal implications of implementing its recommendations, such as the potential for increased resource requirements or the balance between data privacy and utility, etc.

**Claims And Evidence:**

The claims are well-supported by direct quotations and examples from interviews with dataset creators, providing clear evidence of the challenges faced and the efficacy of the recommended practices.

**Datasets And Benchmarks:**

N/A.

**Extended Submissions:**

N/A

**Limitations:**

Although the paper is well written, there are still some limitations to mention about:

* **Representativeness:** Although the data collection tool ranges from various choices (existing ones, crowd work, actors, scaped, etc.), the interview's covered datasets are mainly NLP/text domain datasets.

* **Incentives for the participants**: as specified by authors, "We contacted 47 dataset creators with 18 respondents agreeing to
participate (response rate of 38 percent)". Note that the participants did not receive compensation for their time (> 40 minutes), the incentives for participants to truthfully provide high-quality responses are not that clear. Hence, the trustworthiness of the summarized recommendations may not be fully reliable.

* **Lack of quantitative data**: The paper primarily relies on qualitative insights from interviews with data creators, while quantitative validation of the recommendations' effectiveness is neglected.

**Requested Changes:**

These changes are not required, but I would like to list a few of my concerns/questions while reviewing the paper:

(1) We take the Huggingface datasets for illustration. There are detailed separations of datasets within different AI domains (>30), with around 30 datasets with more than 160K downloads. Since the selected datasets are mainly about NLP/text data, could authors explain the reason for the selected 47 datasets to contact?

(2) In the paper, the usage of quotation marks is not consistent. For example. in the first paragraph of section 3, "dataset creators" v.s. ’dataset creator’.

(3) The caption is not in Figure 1 of the Appendix. Besides, it would be much better if authors could add some illustration text/words for arrows, i.e., what it means to flow from the step "Data-validation" to the step "Parameter and population definitions"; or add some explanations in the caption of figure 1.

(4) In Appendix 2, it would be better if authors could add a brief discussion on why these interview questions are designed, i.e., questions are specifically designed for understanding the challenges of the data collection (question X & question XX), the motivation (question Y & question YY), etc. These would make the paper read more reasonable and clear.

(5) Quantitative analysis: It would be much better if authors could consider incorporating some quantitative analysis to validate the effectiveness of the recommendations.

---

### Review · Reviewer_jajH · 2024-03-04

**Recommendation:** 4
**Confidence:** 3

**Summary Of Contributions:**

While dataset construction is a critical step in machine learning pipelines, it is under-discussed in literature and dataset creators often develop their own best practices in isolation. This paper aims to highlight principles of dataset creation and encourage knowledge sharing by constructing several recommendations based on interviews with creators of well-known machine learning datasets. The paper puts forth seven recommendations touching on topics such as data diversity, quality, documentation, and privacy.

**Strengths:**

1. Recommendations were well-organized and supported with quotations from interviewees, as well as context about the particular datasets and their unique challenges (for example, IKEA and female actors).

2. Beyond discussing what makes a "good" (e.g., high-quality, diverse, not inappropriate, not privacy-violating) dataset, the paper highlights broader aspects of the process. Recommendations made were not only on the static properties of datasets, but also how ML researchers should view the role of datasets (intention in rec 5; searching for existing datasets versus creating new datasets in rec 7; how transparency lines up with incentives in peer-review in rec 4) as well as how we should be mindful of datasets throughout their lifecycle (continuous documentation in rec 4). I appreciated this process-focused framing of datasets and think it makes the recommendations resonate with a broader audience (e.g., of ML researchers/practitioners who do not identify as "data people", but inevitably work with data).

**Broader Impact Concerns:**

None.

**Claims And Evidence:**

All claims/recommendations are well-supported by quotes from conducted interviews.

**Datasets And Benchmarks:**

N/A

**Extended Submissions:**

N/A

**Limitations:**

1. While I understand that the purpose of the paper was to extract general recommendations for responsible dataset design, the 18 datasets studied span various domains, collection strategies, and sizes. Providing more examples from individual datasets or recommendations at the domain/collection strategy/size level could be helpful. For instance, notions of data quality differ across modality (and in multi-modal datasets), and dataset scale can limit the feasibility of certain methods for improving quality/diversity.

2. The changing landscape of machine learning warrants discussion around topics such as synthetic data (using generative text/image models) and dataset contamination during a dataset's lifecycle. This second point is potentially beyond the scope of this paper, as the paper focuses on individual dataset creation, but issues in the broader dataset ecosystem are important and can still influence creation (for example, datasets that have been constructed with hopes of not being in a model's pretraining corpus - things that come to mind are https://arxiv.org/abs/2303.12712 and https://arxiv.org/abs/2307.12375).

3. More details on the interview process could be provided. In particular, why were these interview questions selected? Can you tell us a bit more about the 29 creators who were not interviewed - what was the original composition of datasets you wanted to interview on, versus the 18 you ended up interviewing, and did any of the 29 respond and not agree, or did they all just not respond?

4. The discussion around purpose and use cases of datasets was rather vague, and it isn't clear from the paper what the 18 datasets' use cases were. In particular, while I assume that a lot of datasets are created for simply evaluating a model on a particular domain, a few "meta" use cases come to mind, such as datasets that test for distribution shift (e.g., WILDS, BREEDS) and datasets that test for data curation methods (e.g. DataComp). While these use cases may appear specific, they require certain strategies in dataset design that could be helpful to discuss more broadly. For instance, distribution shift datasets need to have ground-truth annotations of the source and target domains (this could encourage creators to not discard dataset annotations). For research on data curation methods, the paper recommends ensuring that one's dataset is high-quality and free of inappropriate content, but having the pre-filtered versions documented is critical for this line of research. I think these use cases are worth mentioning, especially data curation since it can technically be at odds with some of the recommendations.

**Requested Changes:**

Critical:

* Limitation 1: Can you provide more information on the datasets covered in interviews? For instance, the sizes and a summary from their papers of how they were collected/filtered, what the samples look like, what annotations are provided, what downstream datasets were derived from them? While details around the datasets were supplied when necessary in the body of the paper, having the details all in one place in the Appendix would be helpful in connecting particular interview quotes to the context of the interviewee's dataset.

* Limitation 3: Can you disclose more information about the interviewing process regarding the questions were listed in limitation 3?

Would strengthen the work:

* Limitation 2: Add discussion of synthetic data and data contamination or mention these in the limitations.

* Limitation 4: Provide additional details about the original purposes of each dataset in the study, as well as examples of unintended use cases. Highlight particular use cases beyond better evaluation of a model on a particular task; some use cases that come to mind are datasets for distribution shift and data curation.